# Proteomics and Metabolomics Profiling of Pork Exudate Reveals Meat Spoilage during Storage

**DOI:** 10.3390/metabo12070570

**Published:** 2022-06-21

**Authors:** Fan Zhao, Zhenqian Wei, Yun Bai, Chunbao Li, Guanghong Zhou, Karsten Kristiansen, Chong Wang

**Affiliations:** 1Laboratory of Genomics and Molecular Biomedicine, Department of Biology, University of Copenhagen, 2100 Copenhagen, Denmark; zhao.fan@bio.ku.dk; 2Key Laboratory of Meat Products Processing, Ministry of Agriculture, Jiangsu Collaborative Innovation Center of Meat Production and Processing, Quality and Safety Control, College of Food Science and Technology, Nanjing Agricultural University, Nanjing 210095, China; 2019108080@njau.edu.cn (Z.W.); yunbai@njau.edu.cn (Y.B.); chunbao.li@njau.edu.cn (C.L.); guanghong.zhou@hotmail.com (G.Z.); 3BGI-Shenzhen, Shenzhen 518083, China; 4Institute of Metagenomics, Qingdao-Europe Advanced Institute for Life Sciences, BGI-Qingdao, Qingdao 166555, China

**Keywords:** pork exudate, proteomics, metabolomics, pork quality, temperatures storage, spoilage markers

## Abstract

Previous studies have evaluated pork quality by omics methods. However, proteomics coupled with metabolomics to investigate pork freshness by using pork exudates has not been reported. This study determined the changes in the profiles of peptides and metabolites in exudates from pork stored at different temperatures (25, 10, 4, and −2 °C). Multivariate statistical analysis revealed similar changes in profiles in exudates collected from pork stored at −2 and 4 °C, and additional changes following storage at higher temperatures. We identified peptides from 7 proteins and 30 metabolites differing in abundance between fresh and spoiled pork. Significant correlations between pork quality and most of the peptides from these 7 proteins and 30 metabolites were found. The present study provides insight into changes in the peptide and metabolite profiles of exudates from pork during storage at different temperatures, and our analysis suggests that such changes can be used as markers of pork spoilage.

## 1. Introduction

Meat and meat products represent an important part of the human diet, being rich in protein, vitamins, mineral elements, and other nutrients [1]. The Food and Agriculture Organization (FAO) reported that the global meat production by 2025 is anticipated to be 356 million tons, 16% higher than the baseline period (2013–2015), and is expected to increase to 470 million tons by 2050 [2]. Pork, as the most consumed meat apart from poultry, has been considered an important source of food for humans. The rapid increase in pork demand has led to remarkable developments in the meat industry and contributed to fulfilling the consumers’ rising demands for meat safety, nutrition, and quality [3]. Meat spoilage is a common problem during the different stages of meat production, sale, and storage, causing huge economics losses and concerns in relation to meat safety. The loss attributed to pork spoilage is approximately USD 1 billion per year alone in the USA [4], emphasizing the need for better monitoring of meat spoilage, including the identification of robust markers related to meat spoilage.

Several studies have demonstrated that different storage conditions, such as temperature, affect meat spoilage [5,6]. However, research on meat spoilage has mainly focused on analysis of the meat itself, with little attention being paid to the analysis of the profiles of meat exudates. The loss of exudates from meat is unavoidable during meat storage, caused by loss of water from the muscle tissue [7]. Meat exudate is also released by a disruption of myofibrils, the intracellular cytoskeleton, and cell membranes during storage [8]. Exudates mainly contain water-soluble sarcoplasmic proteins, peptides, amino acids, and low-molecular-mass metabolites [9]. Therefore, meat exudates could be considered as an easily obtained sample enabling analysis to provide information on the entire meat sample.

Recently, studies on meat exudates have reported that the profiles of the composition of exudates may be correlated with meat quality. Thus, García-García et al. (2019) reported that changes in the composition of meat exudates provide valuable information on spoilage during storage, and how these changes are affected by electron beam irradiation [7]. Yu et al. (2021) reported that metabolites detected in pork exudates could act as potential markers to indicate the meat aging process and meat quality [10]. In addition, Xing et al. (2020) observed that more exudate was collected from chickens affected by wooden breast myopathy than from healthy chickens, and that the free hemoglobin concentration and oxidation of lipids were increased [11]. These studies suggest that meat exudates might provide valuable information on meat integrity and that the inspection of changes in exudate profiles could be used as a proxy for meat spoilage.

Mass spectrometry (MS) has been used as an important technology enabling the detection of quantitative and qualitative changes in meat quality due to its wide dynamic range, high resolution, and high sensitivity. Both proteomics- and metabolomics-based MS are effective approaches to monitor changes in composition elicited by endogenous or extrinsic factors by integrating advanced analytical and multivariate statistical methods [12,13]. For instance, Fan et al. (2021) investigated the advantages of super-chilling (−2 °C) storage for maintaining the quality of *Coregonus peled* muscle compared to 0 °C storage, where proteomics analysis based on MS demonstrated that proteins/peptides that differed in abundance between the two groups mainly represented collagen or collagen degradation products [14]. Zhao et al. (2022) reported that there were significant correlations between microorganisms and specific compounds/metabolites detected by liquid chromatography–mass spectrometry (LC–MS), such as small peptides, nucleic acids, and carboxylic acids [15].

The present study aimed to evaluate the feasibility of the LC–MS techniques to determine the protein and metabolite profiles of pork exudate collected from fresh and spoiled pork samples stored at different temperatures, and to examine the correlations between the proteome and metabolome of exudates and pork spoilage.

## 2. Results and Discussion

### 2.1. Identification of Peptides/Proteins by Quantitative Proteomics Analysis

Meat spoilage is mainly the result of the proteolytic activities of different microorganisms or endogenous proteases acting on proteins, converting them into small peptides, which can be detected in exudates. We explored changes in peptide profiles and identified the cognate proteins, comparing exudates from fresh and spoiled samples using a label-free proteomics analysis approach. A total of 4621 peptides were obtained from all groups, which mapped to 195 proteins. The Andromeda score of 86.67% of the peptides was greater than 60, and the median score was 96.02 (Appendix A). Approximately 84.62% of proteins were represented by at least two peptides (Appendix A) and the number of amino acids in the majority of peptides ranged from 8 to 25 (Appendix A). In addition, sequence coverage of cognate proteins greater than 10% and 20% was achieved for 51.79% and 32.82% of the peptides, respectively (Appendix A). These results are comparable with those reported by Wang et al. (2021), who reported a sequence coverage >10% for 56.85% of the analyzed peptides, and a sequence coverage >20% for 39.93% of the analyzed peptides [16]. These results demonstrate that the data used in this study were of good quality.

### 2.2. Identification and Analysis of Differentially Abundant Proteins

To determine to what extent key meat proteins contributed to the changes in peptide profiles, we quantified the abundance of peptides in exudates from control and spoiled samples and calculated the corresponding contributions of the cognate proteins using the MaxQuant software, as detailed in the Materials and Methods. We categorized proteins as differentially abundant proteins (DAPs) if the contributing cognate proteins compared to the control exhibited a fold change ≥2 or ≤0.5 and *p* < 0.05. A total of 35, 37, 95, and 89 DAPs were identified comparing the control to the 25, 10, 4, and −2 °C groups, respectively (Figure 1A). We used volcano plots to visualize the changes in DAPs, where proteins that exhibited an increase in abundance were labeled in red and proteins exhibiting a decreased abundance were labeled in blue. Of note, the number of DAPs in the 10 °C/Control (10/C) and 25 °C/Control (25/C) comparison groups was less than that of the −2 °C/Control (−2/C) and 4 °C/Control (4/C) comparison groups, which may have been due to the stronger degradation of proteins under the relatively high storage temperature [16]. Hierarchical clustering of DAPs using protein quantification of each biological sample was performed (Figure 1B). We observed that the samples separated into two clusters regardless of the storage temperature. The first cluster comprised proteins exhibiting a higher abundance in control samples compared to spoiled samples, and the second cluster comprised proteins exhibiting a higher abundance in the spoiled samples compared to control samples. These results revealed a marked difference in the proteomic patterns of fresh and spoiled samples, indicating that the pattern of proteins was strongly linked to meat spoilage.

To better understand the profiles of DAPs comparing different storage temperatures, we used principal component analysis (PCA) and constructed Venn diagrams. For PCA analysis, the first component explained 52.1% of the total variation and the second component explained 39.8% of the total variation (Figure 1C), indicating that distinct differences in peptide/protein profiles distinguished control and spoiled samples during storage at different temperatures. The short distances between three independent samples in the score plots indicated the reproducibility of the results. The PCA indicated a clear separation between spoiled samples at different temperatures and control samples. However, there was a high degree of similarity between the −2 °C and 4 °C groups. Specifically, the profiles of these two groups were clearly separated from the 10 °C and 25 °C groups on the principal component 1 (PC1) level, indicating the noticeable differences in the changes in peptide/protein components between higher-temperature and lower-temperature storage. These differences between different temperatures may be caused by the activities of various dominant bacteria during meat spoilage. Lawrie and Ledward (2014) summarized that microbes can be classified as psychrotrophs, mesophiles, and thermophiles based on their survival at different temperatures [17]. As seen from Figure 1D, in agreement with the PCA results, the −2 and 4 °C groups exhibited 50 shared DAPs, a much higher number than observed for the high-temperature storage groups. In addition, 5, 10, 11, and 6 DAPs were only found in the 25, 10, 4, and −2 °C groups, respectively; these specific DAPs might be potential biomarkers of the spoilage of pork at these specific temperatures. The seven DAPs (Table 1) that were shared by all groups might represent potential protein biomarkers related to the spoilage of pork during storage at the four different temperatures.

### 2.3. The Metabolic Profile Changes during Pork Spoilage

After peak extraction and alignment, a total of 2978 and 2393 metabolite ion features were detected from all samples in the positive and negative ion modes by ultra-high-performance liquid chromatography–MS/MS (UHPLC–MS/MS)-based non-targeted metabolomics analysis. Based on secondary mass spectrometry and database mapping, a total of 489 and 347 metabolites were annotated. In order to illustrate the differences in metabolites in pork exudates between the control and spoiled samples, we used multivariate statistical analysis methods, including PCA and partial least-squares discriminant analysis (PLS-DA) (Figure 2). The quality control (QC) samples in both modes clustered in the center of the PCA score plot (Figure 2A,B), indicating that the experimental data had high reproducibility and that the monitoring system had good stability. However, in the PCA, some samples of the same group did not cluster tightly, presumably reflecting subtle individual differences between different pork samples. Significant separation was observed between control and spoiled samples. However, the samples of the −2 °C and 4 °C groups could not be distinguished. The PLS-DA scores plot and the validation for pair-wise comparison are displayed in Figure 2C–F. The distribution of the scores plot is consistent with the results of the PCA, and the validation tests confirmed that the PLS-DA models were not overfitted, suggesting the accuracy and validity of the screening of marker metabolites distinguishing the different groups. In addition, the results of the PCA and the PLS-DA were in agreement with the results of the proteomics profiles, indicating that the integrated proteomics and metabolomics approach is useful for investigating differences between fresh and spoiled pork.

To further examine the effect of different temperatures on metabolites during storage, orthogonal partial least-squares discriminant analysis (OPLS-DA) was used to further substantiate the separation of the metabolic profiles between control and spoiled samples. As displayed in Figure 3, all spoiled samples were completely separated from the control samples, further suggesting that the metabolite profiles were significantly affected during storage, regardless of temperature.

### 2.4. Screening of Differentially Abundant Metabolites

The metabolites that differed in abundance between control and spoiled samples were screened based on a combination of Variable Importance in Projection (VIP) ≥1 predicted by OPLS-DA models coupled with fold change ≥2 or ≤0.5 and *p* < 0.05. A total of 99, 65, 88, and 89 and 72, 59, 56, and 65 metabolites showed significantly different abundances in the 25, 10, 4, and −2 °C groups using positive and negative mode, respectively. These differentially abundant metabolites were categorized using Venn diagrams, and the results showed that there were 19 and 18 differentially abundant metabolites that were shared by the four temperature groups in positive (Appendix A) and negative (Appendix A) modes, respectively. Between these 19 and 18 differentially abundant metabolites, seven existed both in positive and negative modes, and thus a total of 30 non-repeated differentially abundant metabolites were screened. The changes in the profiles of these seven shared differentially abundant metabolites between control and spoiled samples in positive and negative modes were highly similar, and the VIP results with higher scores were selected and the corresponding fold changes are listed in Appendix A.

To visualize the profiles of differentially abundant metabolites in the different temperature storage groups, heat map cluster analysis was performed to investigate their abundances and relationship with the storage temperatures (Figure 4A). The heat map revealed that the metabolic profiles were significantly altered in control and spoiled pork. The profile of differentially abundant metabolites in the −2 °C group was similar to that of the 4 °C group. This result is consistent with the PCA of all metabolites. For instance, the abundances of phosphatidylcholine (PC, 34:4), juniperic acid, isopropylmalic acid, and 9, 10-epoxystearic acid were higher in these two groups compared to controls. In addition, specific high-abundance metabolites exhibiting differential abundance could be found at different temperatures, such as dibenzylamine in the 10 °C group, prosopinine and stearolic acid in the 25 °C group, and Ala-Tyr, Val-Val, and carnitine in the control group. As displayed in Appendix A, these differentially abundant metabolites were divided into eight categories, including eight organic/fatty acids, two esters, four nucleotides/nucleosides, five peptides/amino acids, two lipids, two carbohydrates, two alcohols, two ketones, and three other compounds. Nieminen et al. (2016) found metabolites characterized as alcohol, ester, and ketone during pork storage at 4 °C [18]. In addition, Dave et al. (2011) also reported that the accumulation of microbial metabolites, such as aldehydes, alcohols, ketones, esters, amines, organic acids, and sulfur compounds, largely contributed to the spoilage of meat [19].

### 2.5. Analysis of the Correlation between Meat Quality and Omics Data

Several studies have reported that changes in proteins or metabolites during meat storage or post-mortem aging can be used to indicate meat freshness [9,13,20]. Luca et al. (2013) investigated the influence of post-mortem meat aging on the proteome of pork muscle by two-dimensional gel electrophoresis coupled with MS. Three groups of proteins, including stress-related proteins, metabolic enzymes, and structural proteins, exhibited changes in abundance during the post-mortem aging period [21]. In addition, Yu et al. (2021) reported that metabolites associated with ATP synthesis and metabolism, antioxidation, and proteolysis could act as potential markers to evaluate the meat aging process and indicate meat quality [10]. Thus, we constructed a correlation matrix using Pearson’s correlation tests to determine the correlation between meat quality and DAPs, as well as differential metabolites (Figure 4B; the asterisk indicates significant correlation between DAPs/differential metabolites and pork quality characteristics (*p* < 0.05)). The quality characteristics of pork stored at different temperatures were described in our previous study [22]. As seen from Figure 4B, except for the *L** value, most of the DAPs and differentially abundant metabolites exhibited a significant correlation with more than one quality characteristic, which suggests that they could be potential biomarkers for evaluating the quality of pork during storage at different temperatures.

### 2.6. Analysis of Potential Protein Markers Associated with Pork Spoilage

Among these DAPs, LIM domain-binding protein 3 (LDB3), AMP deaminase (AMPD1), and myoglobin (MB) have been reported to be associated with meat spoilage. Li et al. (2014) reported that the content of LDB3 decreases rapidly during refrigerated storage of large yellow croaker, suggesting that the LDB3 protein could be a freshness indicator of large yellow croaker during cold storage [23]. In addition, LDB3 is essential for maintaining Z line structure and muscle integrity [24], and decreased abundance of the LDB3 protein may be related to changes in tenderness after meat spoilage in our study. AMPD1 has been shown to control the generation of inosine 5′-monophosphate (IMP) [25] and the amount of IMP can be used as a freshness indicator of fish products [26]. In addition, England et al. (2015) have demonstrated that the activity of AMPD1 influences the termination of post-mortem glycolysis and the ultimate pH of meat [27]. Myoglobin, as one of the heme proteins, is a marker of the quality of fish muscle, such as color and rancidity development. It plays an important role in several quality parameters, such as whiteness, lipid oxidation, and shelf-life of fish [28]. In addition, Kim et al. (2021) found that the intensity of myoglobin detected by LC–MS proteomics significantly decreased in beef striploin during 21 days of aging [29]. Our results are supported by these studies; even though fish and beef were investigated in their studies, we suggest that these proteins may be indicators of pork freshness during storage.

For other proteins, although they have not been reported to be used as meat freshness indicators, their changes are strongly correlated with meat quality. The elongation factor was degraded significantly with aging time in beef striploin [29] and it can be used as a potential marker reflecting the degree of goat freeze–thaw cycles [30]. In addition, Wu et al. (2014) reported that the degradation of a few large skeletal proteins such as nebulin has been shown to contribute to meat tenderness and other aspects of meat quality [31]. Glycogenin-1 is the protein core involved in the biosynthesis of glycogen. The content of glycogen contributes to post-slaughter changes in pH, which in turn has a great impact on meat quality [32]. Cytochrome c oxidase subunit NDUFA4 is an accessory subunit of cytochrome c oxidase, and Yu et al. (2017) reported that this protein correlated closely with the *a** value of *M. semitendinosus* in Holstein cattle during post-slaughter storage time [33]. Even though these four proteins have not been reported to be associated with meat spoilage, they may be suitable candidates to indicate meat freshness based on our results, but future studies are needed to confirm this suggestion. In addition, it is worth noting that, in our study, the abundance of all these proteins decreased when the meat was spoiled compared to the control; we speculate that the generation and activities of microorganisms may be the main reason for the continuous degradation of these proteins.

### 2.7. Analysis of Potential Metabolite Markers Associated with Pork Spoilage

#### 2.7.1. Organic Acids and Fatty Acids

A study by Argyri et al. (2011) revealed a good correlation between beef spoilage and the amount of organic acids during storage [34]. In our study, juniperic acid and ascorbic acid, as reported previously [20,35], have been associated with food spoilage. Thus, Jia et al. (2019) investigated the meat quality and metabolite profiles of frozen pork tenderloin during 48 h storage and identified 23 differentially abundant metabolites, including juniperic acid [35]. However, they did not investigate the specific correlation between this compound and pork spoilage. Based on our results, finding a higher abundance of juniperic acid in spoiled samples compared to control samples, we speculate that there may be a certain extent of positive association between juniperic acid and pork spoilage. Cheng et al. (2015) found that ascorbic acid was constantly metabolized to threonic acid during the spoilage of cucumber [20]. A reduction in ascorbic acid content and an increase in the microbial population of stored roselle juice with the storage time were reported [36]. Our result is consistent with these studies: the abundance of ascorbic acid was significantly decreased in spoiled samples. In addition, isopropylmalic acid, probably produced by the metabolism of valine, leucine, and isoleucine, has been associated with meat spoilage [37]. This is also consistent with our results, since the abundance of isopropylmalic acid was higher in spoiled samples compared to control samples, while the abundance of the dipeptides Leu-Leu, Leu-Val, and Val-Val decreased in spoiled samples.

#### 2.7.2. Esters

Esters are the dominating components produced by spoilage bacteria during the storage of meat in different conditions. They are formed through the esterification of alcohols and organic acids by microbial esterase activity [38]. Carnitine and tiglylcarnitine can be decomposed into trimethylamine (TMA) through the action of microbes, which contributes to total volatile basic nitrogen (TVB-N) generation during storage [39]. TVB-N is often used as a biomarker of protein and amine degradation, and its content increases with the storage time of meat. In our study, the abundances of carnitine and tiglylcarnitine were significantly decreased in spoiled samples, indicating that they may be converted into TVB-N by microbial action.

#### 2.7.3. Nucleosides and Nucleotides

Nucleosides and nucleotides, particularly adenosine-5′-triphosphate (ATP), have also been shown to contribute to meat freshness. During meat storage, ATP can be degraded by various enzymes into a series of products that deteriorate meat quality, including inosine 5′-monophosphate (IMP), adenosine-5′-diphosphate (ADP), adenosine monophosphate (AMP), hypoxanthine (Hx), and xanthine (Xt). IMP is the main flavor component in meat, but it is unstable and could be further degraded into Hx and inosine. They play an important role in the flavor of meat during storage. Therefore, IMP has become a key indicator of the freshness of meat. As storage time increased, the relative content of IMP dropped dramatically in chilled chicken meat during day 4 to day 6 [40]. Our results are consistent with this study: we found that the abundances of IMP and Hx in spoiled samples were significantly lower than in control samples. In addition, the decrease in the abundance of Hx may be caused by its further degradation to Xt by microbial action. Yu et al. (2021) also reported that Hx declined in pork exudates during aging, and speculated that Hx can be oxidized into Xt via xanthine dehydrogenase [10].

#### 2.7.4. Peptides/Amino Acids

During storage, microorganisms can decompose meat proteins into various intermediate products, such as small peptides and free amino acids [41]. These nutrients can also be used for microbial growth and further metabolized into compounds such as organic acids, sulfur compounds, bioamines, and other substances characteristic of meat spoilage [42]. Liao et al. (2022) reported that amino acid derivatives and oligopeptides derived from protein degradation were the key components to distinguish spoiled and normal hams [43]. Many free amino acids are closely related to meat flavor and can be used as indicators of meat freshness. Gänzle (2015) reported that arginine conversion by lactic acid bacteria in food constitutes spoilage or contributes to flavor [44]. Oligopeptides (Ala-Tyr, Leu-Leu, Leu-Val, and Val-Val) and arginine detected in our study decreased in spoiled samples, suggesting that these compounds have been used by microbes to support their growth. However, Yu et al. (2021) found that the abundance of dipeptides and tripeptides detected from pork exudates increased significantly after 9 d of aging, and suggested that this is caused by proteolysis, which starts after slaughter [10]. The difference from our results may reflect that these peptides are further degraded by microorganisms.

#### 2.7.5. Lipids, Carbohydrates, and Others

Phosphatidylcholine (PC) is a precursor for the production of TMA and has been considered in relation to meat freshness [45]. Chen et al. (2022) showed that the content of PC (18:4/16:1) was significantly increased in salmon muscle stored at 4 °C from the tenth day to the fifteenth day [46]. However, Zhou et al. (2019) found that the content of PC was significantly decreased in mussel (*Mytilus edulis*) meat stored at 4 °C after 4 days of storage [47]. In our study, the abundances of PC (34:4) were significantly increased in spoiled samples in the −2 °C and 4 °C groups, while the abundances decreased in the 10 °C and 25 °C groups. The abundances of PC (36:3) were significantly increased in spoiled samples in all groups. This may be due to the lipid metabolic processes of the various predominant spoilage bacteria at different storage temperatures.

The glucose in raw meat is the first substrate preferably utilized by most of the bacteria growing in meat during storage. When the content of glucose is insufficient, other substances, such as glucose-6-phospate, amino acids, nucleotides, and water-soluble proteins, are metabolized by bacteria [38]. In addition, You et al. (2018) reported that the level of fructose 6-phosphate decreased after 8 days of storage in chilled tan sheep [48]. The abundances of glucose-6-phospate and fructose 6-phosphate were significantly decreased in spoiled samples in our study, indicating that the bacteria have utilized them to support growth. This result is consistent with Zhang et al. (2021), who reported that there is a significant negative correlation between glucose-6-phospate and spoilage bacteria, indicating that glucose-6-phospate may be a substrate for the growth of *Pseudomonasm*, *Serratia*, *Kurthia*, and *Brochothrix* during the spoilage of chilled chicken [3]. Except for the effect of microbial action, several endogenous proteolytic systems presented in meat, including calpains, proteasomes, cathepsins, and other serine peptidases, may also play an important role. For instance, the calpain system is responsible for the majority of the post-mortem degradation of structural proteins, such as myofibrillar proteins, including myosin, tropomyosin, desmin, titin, and nebulin. Furthermore, during the storage of meat in different conditions, the accumulation of microbial metabolites such as alcohols, ketones, and others largely contributed to meat spoilage [38].

A previous study has demonstrated that the storage temperature contributes to the growth of potential spoilage bacteria present in meat [49]. In the present study, the profile changes of peptides from key proteins and metabolites were noted, and subsequent analyses were performed using fresh and spoiled pork stored at four temperatures. This took into account the actual application temperature and possible temperature fluctuations of meat during processing, transportation, and storage. However, since we only investigated four storage temperatures, this is a possible limitation of the study, and future studies employing a wider range of temperatures are warranted to achieve a more comprehensive understanding of the proteins or metabolites produced by spoilage bacteria during storage at specific temperatures.

## 3. Materials and Methods

### 3.1. Sample Collection

A total of 40 *longissimus lumborum* muscles of pork (24 h post-mortem, from 6-month-old crossbred, castrated male pigs (Duroc × Landrace × Large Yorkshire)) were purchased at Beijing Hualian Group (Nanjing, China) and transported to the laboratory on ice within 2 h. After fat and connective tissues were removed, the muscles were cut into 5 × 7 × 3 cm thick steaks (approximately 60 g). Three steaks were assigned and packaged randomly in plastic trays (23 × 14 × 7 cm, Cryovac TQBC-1175, Shanghai, China), wrapped with two layers of commercial polyethylene (PE) cling wrap (Miaojie, TOP Group, Shanghai, China), and placed at different storage temperatures (25, 10, 4, and −2 °C). The exudate produced during meat cutting was collected as fresh pork exudate (control). Exudates from pork stored at 25, 10, 4, and −2 °C were collected with sterilized syringes after 15 h, 72 h, 15 days, and 25 days, respectively, as previously described [22]. The exudates were stored at −80 °C until used for analysis. Three individual steaks were randomly selected for proteomics and metabolomics, respectively.

### 3.2. Proteomics Analysis of Pork Exudate

The pork exudate samples were centrifuged at 12,000× *g* for 20 min at 4 °C, and then the supernatants were transferred into a 10 kDa molecular weight cut-off ultrafiltration tube (Millipore, MA, USA) and centrifuged at 8000× *g* for 10 min at 4 °C. The collected peptides were desalted with C18 cartridges (Sigma-Aldrich, St. Louis, MO, USA). The resulting solution was concentrated by vacuum centrifugation and reconstituted in 40 μL of 0.1% (*v*/*v*) trifluoroacetic acid solution for liquid chromatography–tandem mass spectrometry (LC–MS/MS) analysis.

A Nano-LC tandem with Q-Exactive mass spectrometer (Thermo Fisher Scientific, Waltham, USA) was used to perform peptide analysis. First, 5 μg of peptides were loaded onto a C18 column (10 cm × 75 µm × 3 µm, Thermo Fisher Scientific, Waltham, MA, USA) and separated with a linear gradient of buffer A (2% acetonitrile and 0.1% formic acid) and buffer B (80% acetonitrile and 0.1% formic acid) at a flow rate of 250 nL/min, controlled by IntelliFlow technology over 60 min. MS data were acquired using a data-dependent top 10 method, dynamically choosing the most abundant precursor ions from the survey scan (300–1800 *m*/*z*) for higher-energy collisional dissociation (HCD) fragmentation. The dynamic exclusion duration was 25 s. Survey scans were acquired at a resolution of 70,000 at *m*/*z* 200 and the resolution for HCD spectra was set to 17,500 at *m*/*z* 200.

The data were analyzed using MaxQuant software version 1.5.5.1 and searched against the UniProt *Sus scrofa* database (49,792 total entries, downloaded on 16 August 2021). The precursor mass and MS/MS tolerance of peptide were set to 6 and 20 ppm, respectively. No enzyme was selected to conduct enzymatic cleavage. The cut-off of global false discovery rate (FDR) for peptide and protein identification was set to 0.01. The protein intensity was quantified based on the razor and unique peptide intensity; the razor peptide can only contribute to the identification score of the protein group that has the highest probability. Differentially abundant proteins (DAPs) were characterized as proteins with a fold change in intensity ≥2 or ≤0.5 and *p* < 0.05.

### 3.3. Metabolomics Analysis of Pork Exudate

The samples were centrifuged at 15,000× *g* for 15 min at 4 °C. The supernatants (100 μL) were mixed with 400 μL of cold methanol/acetonitrile (1:1, *v*/*v*) to remove protein and centrifuged at 15,000× *g* for 15 min at 4 °C. The supernatants were withdrawn and dried in a vacuum centrifuge and re-dissolved in 100 μL of acetonitrile/water (1:1, *v*/*v*) solvent. To monitor the stability and reproducibility of instrument analysis, quality control (QC) samples were prepared by pooling 10 μL of each sample and analyzing these pooled samples together with the other samples. The QC samples were analyzed for every 5 samples.

An UltiMate 3000 HPLC tandem with Q-Exactive mass spectrometer (Thermo Fisher Scientific, Waltham, USA) was used to perform the metabolomics analyses. First, 10 μL of sample was loaded onto an ACQUIY UPLC BEH column (2.1 mm × 100 mm × 1.7 µm, Waters, Ireland). The mobile phases consisted of solvents A (25 mM ammonium acetate and 25 mM ammonium hydroxide in water) and B (acetonitrile). The gradient was 85% B for 1 min, linearly reduced to 65% in 11 min, and then reduced to 40% in 0.1 min and kept for 4 min, and then increased to 85% in 0.1 min, with a 5 min re-equilibration period employed. The positive and negative electrospray ionization modes were applied and conditions were as follows: ion source gas 1 as 60, ion source gas 2 as 60, curtain gas as 30, source temperature of 600 °C. In MS-only acquisition, the instrument was set to acquire data over the *m*/*z* range 60–1000 Da, and the accumulation time for time-of-flight (TOF) MS scan was set at 0.20 s/spectra. In auto MS/MS acquisition, the instrument was set to acquire data over the *m*/*z* range 25–1000 Da, and the accumulation time for product ion scan was set at 0.05 s/spectra.

The raw MS data (RAW files) were converted to MzXML files using ProteoWizard MSConvert before importing them into the freely available XCMS software. For peak selection, the following parameters were used: centwave *m*/*z* = 25 ppm, peakwidth = c (10, 60), prefilter = c (10, 100). For peak grouping, bw = 5, mzwid = 0.025, minfrac = 0.5 were used. Collection of algorithms of metabolite profile annotation was used for the annotation of isotopes and adducts. In the extracted ion features, only the variables having more than 50% of the non-zero measurement values in at least one group were kept. Compound identification of metabolites was performed by comparing MS/MS spectra with an accuracy *m*/*z* value < 25 ppm with an in-house database established with available authentic standards.

### 3.4. Multivariate Statistical Analysis of Metabolites

Multivariate data analysis was performed using SIMCA-P (version 17.0, Umetrics, Umea, Sweden). Pareto-scaled principal component analysis (PCA), partial least-squares discriminant analysis (PLS-DA), and orthogonal partial least-squares discriminant analysis (OPLS-DA) were executed to classify and discriminate the difference between the different groups. Seven-fold cross-validation and response permutation testing (200) were used to evaluate the robustness of the model. The variable importance in the projection (VIP) value of each variable in the OPLS-DA model was calculated. The screening criterion of differential metabolites was a VIP value >1, Student’s *t*-test < 0.05, and the fold change ≥2 or ≤0.5 between spoiled and control samples. The Spearman’s correlation coefficient between the biological parameters of meat quality and screened DAPs as well as differential metabolites was analyzed using the R package.

## 4. Conclusions

In the present study, the changes in the profiles of proteins matching identified peptides and metabolites in exudates from fresh and spoiled pork stored at different temperatures were investigated using proteomics and metabolomics. A total of seven DAPs including LDB3, AMPD1, and MB and 30 metabolites including organic acids, fatty acids, esters, nucleotides/nucleosides, peptides/amino acids, and ketones were screened as potential biomarkers to indicate pork spoilage regardless of storage temperature. The combined analysis of data compiled by proteomics and metabolomics indicated that the changes in the profiles of peptides/proteins and metabolites in pork stored at −2 °C and 4 °C were similar. In addition, the screened DAPs and differentially abundant metabolites were significantly correlated with pork quality. This study provides insight into processes associated with pork spoilage at different temperatures and describes new detection methods to determine pork freshness by analyzing pork exudates.

## Figures and Tables

**Figure 1 metabolites-12-00570-f001:**
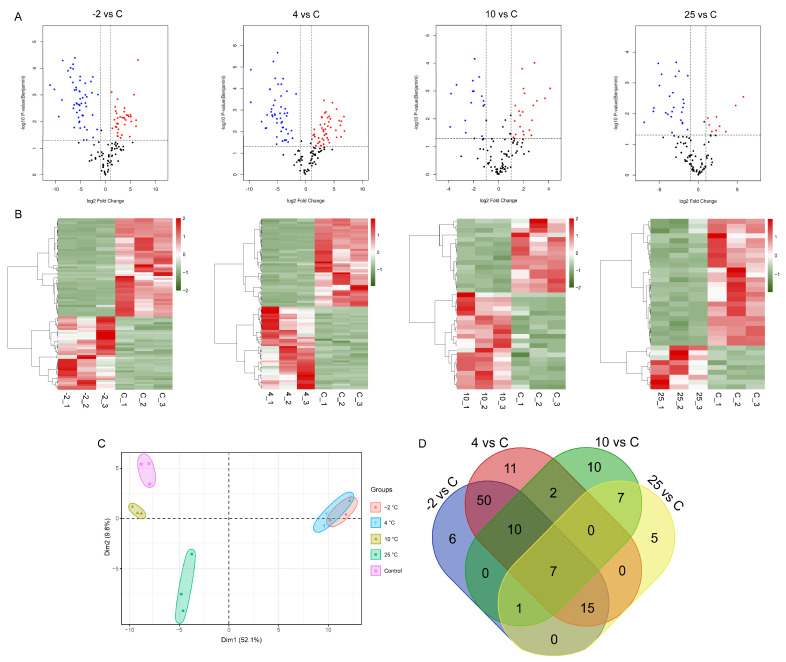
Volcano plots (**A**), hierarchical cluster analysis (**B**), principal component analysis (**C**), and Venn diagrams (**D**) of differentially abundant proteins (DAPs) in exudates from pork stored at different temperatures. Pork exudates were collected from fresh and spoiled pork stored at 25, 10, 4, and −2 °C for 15 h, 72 h, 15 days, and 25 days, respectively. Here, −2, 4, 10, and 25 refer to the storage temperature. C refers to control (fresh pork exudate). The values on the Venn diagrams refer to the number of identified DAPs.

**Figure 2 metabolites-12-00570-f002:**
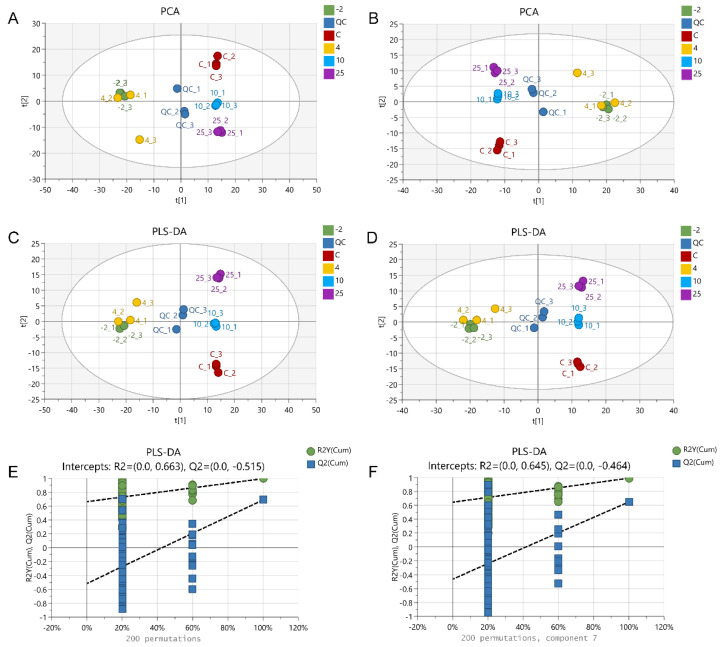
PCA score plots (**A**,**B**), PLS-DA score plots (**C**,**D**), and permutation validation plot (**E**,**F**) of identified metabolites in exudates from pork stored at different temperatures. Pork exudates were collected from fresh and spoiled pork stored at 25, 10, 4, and −2 °C for 15 h, 72 h, 15 days, and 25 days, respectively. Here, −2, 4, 10, and 25 refer to the storage temperature. C refers to control (fresh pork exudate). QC refers to quality control samples prepared by pooling each sample.

**Figure 3 metabolites-12-00570-f003:**
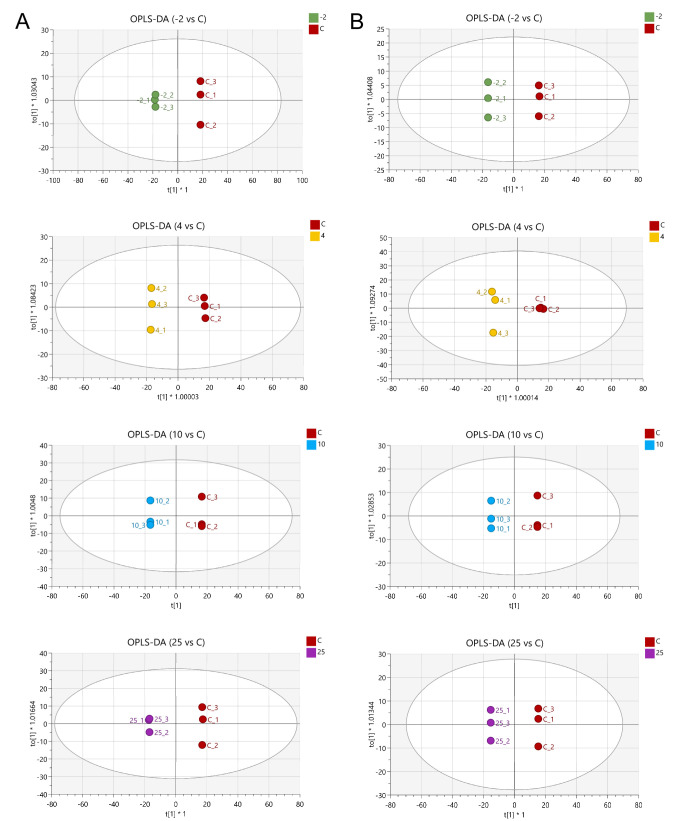
OPLS-DA score plots of identified metabolites in exudates from pork stored at different temperatures. (**A**) OPLS-DA analysis of identified metabolites in pork exudate using positive mode; (**B**) OPLS-DA analysis of identified metabolites in pork exudate using negative mode. Pork exudates were collected from fresh and spoiled pork stored at 25, 10, 4, and −2 °C for 15 h, 72 h, 15 days, and 25 days, respectively. Here, −2, 4, 10, and 25 refer to the storage temperature. C refers to control (fresh pork exudate).

**Figure 4 metabolites-12-00570-f004:**
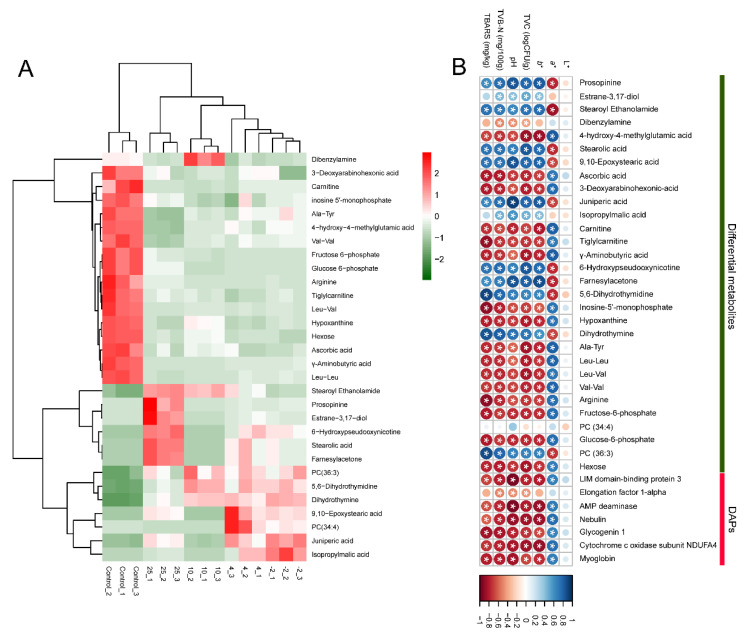
Hierarchical cluster analysis of differentially abundant metabolites in exudates from pork stored at different temperatures (**A**) and correlation between the DAPs/differentially abundant metabolites and meat quality (**B**). Pork exudates were collected from fresh and spoiled pork stored at 25, 10, 4, and −2 °C for 15 h, 72 h, 15 days, and 25 days, respectively. Here, −2, 4, 10, and 25 refer to the storage temperature. C refers to control (fresh pork exudate). Blue color in (**B**) indicates a positive correlation and red color indicates a negative correlation. Asterisk indicates *p* < 0.05. TVC refers to total viable count, TVB-N refers to total volatile basic nitrogen, and TBARS refers to thiobarbituric acid reactive substances. *L**, *a**, and *b** refer to the lightness, redness, and yellowness of pork, respectively.

**Table 1 metabolites-12-00570-t001:** Shared differentially abundant proteins (DAPs) in exudates from pork stored at different temperatures compared to fresh meat.

Accession No.	Protein Name	Gene Name	Peptide Counts	Mol. Weight [kDa]	Ratio (*p* < 0.05)			
					−2 vs. C	4 vs. C	10 vs. C	25 vs. C
A0A287A435	LIM domain-binding protein 3	LDB3	207	65.603	0.015	0.019	0.342	0.090
A0A288CG57	Elongation factor 1-alpha	EEF1A1	14	49.07	0.016	0.013	0.493	0.143
B5SYT7	AMP deaminase	AMPD1	21	86.502	0.005	0.013	0.267	0.021
F1SHX0	Nebulin	NEB	358	772.72	0.070	0.102	0.420	0.030
F1SKC4	Glycogenin 1	GYG1	21	37.309	0.001	0.001	0.441	0.151
I3LJI1	Cytochrome c oxidase subunit NDUFA4	NDUFA4	2	15.368	0.012	0.023	0.098	0.041
P02189	Myoglobin	MB	47	17.084	0.137	0.087	0.401	0.251

## Data Availability

The peptidomics and metabolomics datasets generated for this study can be found in ProteomeXchange, Accession No. PXD033970.

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
