# Peer review of "Proteomics and Metabolomics Profiling of Pork Exudate Reveals Meat Spoilage during Storage"

_metabolites, 2022, doi:10.3390/metabo12070570_

Round 1

Reviewer 1 Report

The article “Proteomics and metabolomics profiling of pork exudate reveals meat spoilage during storage” is written in scientific, correct English. The problems being addressed in manuscript are also potentially of interest to readers of Metabolites. The introduction and discussion are explanatory and I have no objection to the methodologies.  Summarizing, I found the article interesting and the subject worthy to be published in present form.

Please only change Keywords to: pork exudate; proteomics; metabolomics; pork quality; storage temperature; spoilage markers

Author Response

The article “Proteomics and metabolomics profiling of pork exudate reveals meat spoilage during storage” is written in scientific, correct English. The problems being addressed in manuscript are also potentially of interest to readers of Metabolites. The introduction and discussion are explanatory and I have no objection to the methodologies.  Summarizing, I found the article interesting and the subject worthy to be published in present form.

Please only change Keywords to: pork exudate; proteomics; metabolomics; pork quality; storage temperature; spoilage markers

Response. We appreciate the very positive comments. We have revised our manuscript according to your comments.

Reviewer 2 Report

1  In chapter 2.1. the role of bacteria in the conversion of meat proteins into peptides is reported. The authors do not mention the role of endogenous proteases (calpains, cathepsins), whose role in protein degradation is indisputable. This is probably the case with nebulin (Table 1).

2The manuscript lacks a list of abbreviations, resp. many abbreviations used in the text or in the descriptions of the figures are not explained at all.

3  Figure 4B contains CIELab or TVC or TBARS values, but there is no mention of these analyzes in Chapter 3. There is no reference to another study by the authors where they would deal with these analyzes.

4If in Figure 4B the color red means a positive correlation, then I do not understand the relationship between TVC and the content of eg carnitine, where according to lines 283 and 284 there is a negative correlation, but the color between TVC and carnitine is red. At the same time, the number of TVCs increases with deepening spoilage. The same color mismatch is found for IMP, hexose or G-6-P or juniperic acid. A similar relationship applies between the pH value (it increases when meat is spoiled) and the mentioned substances in the meat. Table S1 does not provide a satisfactory explanation.

Conversely, as the TVB-N concentration increases, the carnitine concentration decreases. I.e. negative correlation and blue color. But in figure 4B it is red!

5  In chapter 1 on line 33 of the mentioned 200 million tons of meat production is low, correctly according to the FAO 356 million tons.

6Minor inaccuracies in the text: eg line 362: at 12,000 g, but on line 389: at 15000 g. It is good to unify the form of notation.

Author Response

  1. In chapter 2.1. the role of bacteria in the conversion of meat proteins into peptides is reported. The authors do not mention the role of endogenous proteases (calpains, cathepsins), whose role in protein degradation is indisputable. This is probably the case with nebulin (Table 1).

Response. We have discussed the role of endogenous proteases (calpains, proteasomes, cathepsins, and other serine peptidases) in protein degradation in Line 364-369, revising our manuscript according to your comments.

  1. The manuscript lacks a list of abbreviations, resp. many abbreviations used in the text or in the descriptions of the figures are not explained at all.

Response. We have added the full name of abbreviations according to your comments.

  1. Figure 4B contains CIELab or TVC or TBARS values, but there is no mention of these analyzes in Chapter 3. There is no reference to another study by the authors where they would deal with these analyzes.

Response. The quality characteristics (L*, a*, b*, TVC, pH, TVB-N and TBARS) of pork stored at different temperatures were described and analyzed in our previous study and referred to as reference 22.

  1. If in Figure 4B the color red means a positive correlation, then I do not understand the relationship between TVC and the content of eg carnitine, where according to lines 283 and 284 there is a negative correlation, but the color between TVC and carnitine is red. At the same time, the number of TVCs increases with deepening spoilage. The same color mismatch is found for IMP, hexose or G-6-P or juniperic acid. A similar relationship applies between the pH value (it increases when meat is spoiled) and the mentioned substances in the meat. Table S1 does not provide a satisfactory explanation.

Conversely, as the TVB-N concentration increases, the carnitine concentration decreases. I.e. negative correlation and blue color. But in figure 4B it is red!

Response. We are sorry that we described the relationship incorrectly. It can be seen from figure 4B, the color blue means a positive correlation since the value in the bottom of figure 4B is 0 to 1, and the color red means a negative correlation since the value is 0 to -1. We have revised this description.

  1. In chapter 1 on line 33 of the mentioned 200 million tons of meat production is low, correctly according to the FAO 356 million tons.

Response. We have revised this value according to your comments.

  1. Minor inaccuracies in the text: eg line 362: at 12,000 g, but on line 389: at 15000 g. It is good to unify the form of notation.

Response. We have corrected the inaccuracies in our manuscript according to your comments.

Reviewer 3 Report

Title: Proteomics and metabolomics profiling of pork exudate reveals 2 meat spoilage during storage

The manuscript “Proteomics and metabolomics profiling of pork exudate reveals 2 meat spoilage during storage” is well written, however, the major concerns are:

Lack of novelty. This reviewer noticed one difference from previous studies. Previous studies used different techniques than current study.

·       https://doi.org/10.1186/1477-5956-11-9

·       https://doi.org/10.3390/foods10030668

Authors did not cite or acknowledge these researches. Please include and discuss the results of these published papers.

Line 53-60: Better to give references related to the pork exudate as there are research articles available in the literature.

The discussion section is very weak. The authors need to explain insight of correlation between meat quality and omics data by providing some relevant references. More relevant research articles need to be incorporated in order to elaborate the potential protein and metabolite markers.

Authors should provide more information about the collection of the samples?

Please mention the sample size in the materials and methods part.

Another major concern is that the study was designed on the meat exudate, however, meat was purchased from local market, of which authors don’t know the postmortem time at which experiment was performed and other handling procedure. Why authors don’t get meat from slaughterhouse under control time and conditions.

How the data was normalized, did authors used any statistical technique such as log transformation and pareto Scaling before multivariate analysis? Please mention in the materials and methods part.

It would be better to have Kyoto Encyclopedia of Genes and Genomes (KEGG) analysis using differential metabolites, as it will be easier for the readers to understand different biological processes involved in spoilage of pork meat.

Some of the samples PCA results were not tightly clustered, authors should explain the reason about these outliers, within manuscript.

English needs to be thoroughly improved in the manuscript, grammar, syntax and style errors need to be revised.

Author Response

The manuscript “Proteomics and metabolomics profiling of pork exudate reveals 2 meat spoilage during storage” is well written, however, the major concerns are:

Lack of novelty. This reviewer noticed one difference from previous studies. Previous studies used different techniques than current study.

  • https://doi.org/10.1186/1477-5956-11-9
  • https://doi.org/10.3390/foods10030668

Authors did not cite or acknowledge these researches. Please include and discuss the results of these published papers.

Response. Thank you very much for drawing our attention to these references. We have included and discussed the results of these articles in our manuscript (Line 230- 237, Line 324-326, Line 339-342).

Line 53-60: Better to give references related to the pork exudate as there are research articles available in the literature.

Response. Reference 7 concerns a study related to the pork exudates, but we have added an additional reference (10).

The discussion section is very weak. The authors need to explain insight of correlation between meat quality and omics data by providing some relevant references. More relevant research articles need to be incorporated in order to elaborate the potential protein and metabolite markers.

Response. We have revised our manuscript according to your comments (Line 251-253, Line 255-257, Line 297-301).

Authors should provide more information about the collection of the samples?

Response. We have revised our manuscript according to your comments (Line 384-396).

Please mention the sample size in the materials and methods part.

Response. We have included information on the sample size according to your comments.

Another major concern is that the study was designed on the meat exudate, however, meat was purchased from local market, of which authors don’t know the postmortem time at which experiment was performed and other handling procedure. Why authors don’t get meat from slaughterhouse under control time and conditions.

Response. In fact, the samples we studied were obtained from the slaughterhouse, but not directly. We entrusted Beijing Hualian Group to help us complete the transportation from the slaughterhouse to their store. So, we have detailed information about the meat samples and the information were provided in our previous study (reference 22).

How the data was normalized, did authors used any statistical technique such as log transformation and pareto Scaling before multivariate analysis? Please mention in the materials and methods part.

Response. When we analyzed the fold change of differentially abundant proteins and differentially abundant metabolites, the original intensity was used, and normalization was not applied.

It would be better to have Kyoto Encyclopedia of Genes and Genomes (KEGG) analysis using differential metabolites, as it will be easier for the readers to understand different biological processes involved in spoilage of pork meat.

Response. At first, we tried to use KEGG to investigate the biological processes involved in the generation of metabolites exhibiting differential abundances, but due to limitations related to the species (Sus scrofa), very little information is available and therefore we did not include these incomplete results.

Some of the samples PCA results were not tightly clustered, authors should explain the reason about these outliers, within manuscript.

Response. We have explained the possible reason about this result in our manuscript (Line 157-159).

English needs to be thoroughly improved in the manuscript, grammar, syntax and style errors need to be revised.

Response. We have carefully revised the manuscript.

Round 2

Reviewer 2 Report

Thank you for revising the manuscript.The authors have performed the required revision of the manuscript, which can now be published.

Reviewer 3 Report

Accept in present form.